# MYOC Promotes the Differentiation of C2C12 Cells by Regulation of the TGF-β Signaling Pathways via CAV1

**DOI:** 10.3390/biology10070686

**Published:** 2021-07-20

**Authors:** Yuhan Zhang, Shuang Li, Xin Wen, Huili Tong, Shufeng Li, Yunqin Yan

**Affiliations:** The Laboratory of Cell and Development, Northeast Agricultural University, Harbin 150030, China; z18845635247@163.com (Y.Z.); lishuang8927@neau.edu.cn (S.L.); wygsygxts@163.com (X.W.); tonghuili@neau.edu.cn (H.T.); lishufeng1@neau.edu.cn (S.L.)

**Keywords:** C2C12, differentiation, MYOC, CAV1, TGF-β

## Abstract

**Simple Summary:**

MYOC is a secreted glycoprotein and it expresses at high levels in skeletal muscle cells. However, the function of MYOC in muscle is still unclear. Accordingly, in this study, we examined that MYOC expression increased gradually during C2C12 differentiation and it could promote the differentiation of C2C12. Furthermore, we demonstrated that MYOC could bind to CAV1. We further confirmed that CAV1 could positively regulate C2C12 differentiation through the TGF-β pathway. At last, we determined the relationship among MYOC, CAV1 and TGF-β. We found that MYOC promoted the differentiation of C2C12 cells by regulation of the TGF-β signaling pathways via CAV1. The present study is the first to demonstrate the mechanism of action of MYOC in C2C12 cells. It provides a novel method of exploring the mechanism of muscle differentiation and represents a potential novel method for the treatment of muscle diseases.

**Abstract:**

Myocilin (MYOC) is a glycoprotein encoded by a gene associated with glaucoma pathology. In addition to the eyes, it also expresses at high transcription levels in the heart and skeletal muscle. MYOC affects the formation of the murine gastrocnemius muscle and is associated with the differentiation of mouse osteoblasts, but its role in the differentiation of C2C12 cells has not yet been reported. Here, MYOC expression was found to increase gradually during the differentiation of C2C12 cells. Overexpression of MYOC resulted in enhanced differentiation of C2C12 cells while its inhibition caused reduced differentiation. Furthermore, immunoprecipitation indicated that MYOC binds to Caveolin-1 (CAV1), a protein that influences the TGF-β pathway. Laser confocal microscopy also revealed the common sites of action of the two during the differentiation of C2C12 cells. Additionally, CAV1 was upregulated significantly as C2C12 cells differentiated, with CAV1 able to influence the differentiation of the cells. Furthermore, the Western blotting analysis demonstrated that the expression of MYOC affected the TGF-β pathway. Finally, MYOC was overexpressed while CAV1 was inhibited. The results indicate that reduced CAV1 expression blocked the promotion of C2C12 cell differentiation by MYOC. In conclusion, the results demonstrated that MYOC regulates TGF-β by influencing CAV1 to promote the differentiation of C2C12 cells.

## 1. Introduction

Skeletal muscle is essential for movement, postural support, breathing, and heat generation [1,2]. Its growth and development is a complex process, requiring muscle stem cells to differentiate into mononuclear myoblasts, the fusion of myoblasts into multinucleated myotubes, and the maturation of muscle fibers [3,4,5]. C2C12 cells are skeletal muscle-derived myoblasts and are widely used to model metabolic diseases and muscle injury repair processes [6]. The study of C2C12 cell differentiation is crucial for drug development and elucidation of the mechanisms of muscle-related diseases.

The differentiation of C1C12 cells is regulated by multiple factors. The myogenic regulatory factors (MRFs) myogenic factor 5 (Myf5), myogenic differentiation antigen (MyoD), myogenin (MYOG), and myogenic factor 4 (MRF4) are members of the basic helix-loop-helix family of transcription factors that control the differentiation of skeletal muscle cells [7,8]. Myosin heavy chains (MyHCs) are key factors that affect muscle function, growth, and development [9]. Desmin, a member of the intermediate filament (IF) family, is muscle-specific and a known myogenic marker in skeletal muscle [10].

The proliferation and differentiation of C2C12 cells are also regulated by multiple signaling pathways, including mTOR, Notch, transforming growth factor (TGF-β), etc. [11,12,13]. The TGF-β superfamily consists of a group of polypeptide cytokines that act by binding TGF-β factors to TGF-β receptors (TβRs) located on cell membranes. TβRs, including type Ι (TβR-Ι), type II (TβR-II), and type III (TβR-III) allow extracellular information to be passed into cells, thereby regulating the expression of Smad2/3, controlling the translocation of Smad4 into the nucleus which regulates the differentiation, proliferation, apoptosis, and migratory behavior of cells [14,15,16,17].

Caveolin-1 (CAV1) is a 22 kDa membrane protein essential for the formation of small indentations (fossa) in the plasma membrane. It regulates the activity of receptors or the assembly of signaling molecules through scaffold sequences, promoting their function and so participating in signal transduction [18]. The CAV1 gene is principally expressed in endothelial cells, smooth muscle cells, and fibroblasts [19]. Activin receptor-like kinase (AKL) is a protein that is a type of TβR-Ι. Studies have shown that CAV1 binds to AKL, thereby activating the TGF-β pathway in mouse embryonic fibroblasts and rat myoblasts [18,20].

The glycoprotein myocilin (MYOC) and its associated gene have been extensively studied in open-angle glaucoma, but studies in other cells have rarely been reported [21]. Studies have shown that MYOC affects the formation of the murine gastrocnemius muscle. MYOC also plays an important role in the differentiation of mouse osteoblasts both in vitro and in vivo [22,23]. Additionally, MYOC promotes cell proliferation and resistance to apoptosis via the ERK1/2 MAPK signaling pathway [24]. However, the expression and mechanism of action of MYOC in the differentiation of C2C12 cells remain unreported.

Here, we aim to investigate the effect of MYOC on the differentiation of C2C12 cells and explore the mechanism by which MYOC operates during C2C12 differentiation. The study verified that MYOC regulates the TGF-β pathway through CAV1 to promote the differentiation of C2C12 cells.

## 2. Materials and Methods

### 2.1. Cell Culture and Differentiation

The C2C12 myoblasts used in our laboratory were donated by Professor Guangpeng Li from Inner Mongolia University. C2C12 myoblasts were cultured in high glucose Dulbecco’s modified Eagle medium (DMEM, Gibco, New York, NY, USA) supplemented with 10% fetal bovine serum (FBS) and 0.5% penicillin/streptomycin, at 37 °C in humid air containing 5% CO_2_. The cells were routinely passaged when 70% confluent. The cells were seeded into 6-well culture plates. When the cells were 85–90% confluent, the FBS was replaced by a 2% horse serum (Gibco)-containing a differentiation medium to induce differentiation of the C2C12 myoblasts.

### 2.2. Western Blotting

C2C12 cells were cultured in six-well plates and then washed with phosphate-buffered saline (PBS) cooled to 4 °C twice, after which RIPA cell lysis buffer (Beyotime, Shanghai, China) was added prior to collection of total cellular protein. The protein was denatured by heating the sample in boiling water for 10 min. Each lane of a sodium dodecyl sulfate (SDS)-polyacrylamide gel was loaded with approximately 20 μL of extracted protein. Used 10% separation glue and 5% concentrate glue electrophoresed, after which the protein bands were transferred to polyvinylidene fluoride (PVDF) membranes (Millipore Corporation, Boston, MA, USA). The PVDF membranes were placed in a blocking buffer (PBS with 5% skimmed milk) then incubated at 37 °C for 1 h. The same buffer was used to dilute the primary and secondary antibodies. Each PVDF membrane was incubated with primary and secondary antibodies sequentially. Finally, protein blots were developed using a Super ECL Plus kit (Applygen Technologies, Inc., Beijing, China) and visualized using a small chemiluminescence imager (Sage Creation, Beijing, China). All Western blotting raw data in this paper are shown in Appendix A. Antibodies used for Western blotting were: anti-MYOC (1:1000, Sangon, D154098, Shanghai, China), anti-CAV1 (1:1000, Sangon, D261423), anti-MYH2 (1:1000, Santa Cruz Biotechnology, SC-53092, Dallas, TX, USA), anti-MYOG (1:500, Santa, SC-12732), anti-GAPDH (1:2000, Proteintech, 10494-1-AP, Rosemont, IL, USA), anti-Smad2 (1:1000, bs-0718R, Bioss, Beijing, China), anti-P-Smad2 (1:500, bs-7464R, Bioss), anti-Smad4 (1:1000, D120124, Sangon), and anti-P-Smad4 (1:500, Affinity, Ab-AF8316, Jiangsu, China).

### 2.3. Immunofluorescence and Laser Confocal Microscopy

Cells cultured in 6-well plates were rinsed 3 times with PBS then fixed at −20 °C for 20 min after the addition of pre-cooled methanol. The methanol was discarded and the cells were rinsed with PBST (PBS with 0.5% Triton X-100). The cells were then incubated at 37 °C with 5% bovine serum albumin (BSA; Biotopped, Beijing, China) blocking buffer in PBST for 1 h. Primary antibody (anti-MYOC, anti-CAV1, or anti-Desmin Bioss) diluted 1:50 with blocking buffer was added then incubated overnight at 4 °C. Secondary antibody fluorescently labeled with either FITC (Bioss) or RBFITC (Bioss), diluted with blocking buffer, was added and incubated at 37 °C for 2 h. Cell nuclei were stained with 200 μL DAPI (Thermo Fisher Scientific, Shanghai, China) at room temperature for 5 min. Two drops of an anti-fluorescence quenching agent (Beyotime) were added to the layer of cells for sealing. Fluorescent images were acquired using an inverted fluorescence microscope (Olympus, BX43, Tokyo, Japan) and a laser confocal microscope (Leica, SP8 AOBS, Wetzlar, Germany). To have better effects of scanning and photography, we used the laser confocal microscope with a 40× oil immersion lens, turning on Z-stack mode and adjusting the moving microscope table to find the layer with the clearest vision.

The myotube fusion index was calculated after the cells were stained with Desmin from the number of cells that had fused into myotubules/the total number of cells using the mean values in five fields of vision from each experiment.

### 2.4. Construction of MYOC Overexpression Vector

The following primer sequences for MYOC cDNA were synthesized as follows:

Upstream primer: 5′-CGCGGATCCCAGGAGAACTTTCCAGAAGAAACC-3′;

Downstream primer: 5′-GCTCTAGACAGAAGAGAACAGAGTCTATGCGA-3′ (Sangon). Full-length MYOC cDNA was obtained by PCR. The cDNA sequence of MYOC was attached to pcDNA3.1(+) (Beyotime) and then sequenced to ensure it was correct (Sangon). The constructed overexpression plasmid was termed pcDNA3.1(+)-MYOC. The plasmid was extracted with an extraction kit (Sangon), and its concentration was measured.

### 2.5. Transfection of Overexpression Vector and siRNA

The overexpression plasmid was transfected into C2C12 cells using polyetherimide (PEI; Sigma-Aldrich, St. Louis, MO, USA). Three MYOC siRNA sequences were purchased, termed T1, T2, and T3. After screening, T3 was used for subsequent experiments (Appendix A): Upstream primer for T3: 5′-CAAAGGAUGUGGAGCGCUATT-3′; Downstream primer for T3: 5′-UAGCGCUCCACAUCCUUUGTT-3′ (Sangon). Three siRNA sequences for CAV1, termed S1, S2, and S3 (Sangon Biotech) were synthesized. After screening, S1 was identified for use in subsequent experiments (Appendix A): Upstream primer for S1: 5′-GGCAAGAUAUUCAGCAACATT-3′; Downstream primer for S1: 5′-UGUUGCUGAAUAUCUUGCCTT-3′ (Sangon). C2C12 cells were transfected with siRNA using Lipofectamine 2000 (Invitrogen, Carlsbad, CA, USA) in accordance with the manufacturer’s recommended protocol.

### 2.6. Co-Immunoprecipitation

C2C12 cells were collected on the 7th day after differentiation. The cells were washed with PBS three times then lyzed with 1 mL RIPA cell lysis buffer for 2 h at 4 °C. A 20 μL aliquot of protein extract was removed and used as the input sample. The remaining protein sample was then divided into two equal parts, into which either 2 μg Rabbit IgG (Beyotime, A7016) or anti-MYOC antibody were added, respectively, which were then incubated overnight at 4 °C. Forty μL of protein A + G agarose beads (Beyotime, P2012) were added to each of the two samples and incubated for 4 h at 4 °C. After centrifugation, the protein samples were washed with buffer 5 times, and the harvested beads were resuspended in 5× SDS sample buffer then boiled for 10 min. Each protein precipitate isolated by Western blotting was stained with Coomassie brilliant blue. Specific bands were then sequenced (Sangon).

### 2.7. Statistical Analysis

Each set of experiments was repeated at least three times, the mean value representing each experimental result. Grayscale scanning of the protein bands on Western blots was performed using ImageJ software (National Institutes of Health, Bethesda, MD, USA), and the resultant data was analyzed using GraphPad Prism (GraphPad Software, La Jolla, CA, USA). All data are expressed as means ± standard deviations. Statistical analyses were performed using SPSS software (IBM Corp., Armonk, NY, USA). *t*-tests were used to analyze variance, while a post hoc test was used for comparisons of multiple groups. *p* values < 0.05 indicated statistical significance.

## 3. Results

### 3.1. Expression Pattern and Localization of MYOC during the Differentiation of C2C12 Cells

To evaluate the expression levels of MYOC at different stages of C2C12 cell differentiation, C2C12 cells at different stages (0, 1, 3, 5, and 7 days after the differentiation of C2C12 cells) were collected. The Western blotting results indicated that the expression of MYOC increased significantly upon C2C12 cell differentiation (Figure 1A,B). Simultaneously, the gene expression levels of the differentiation markers MYH2 and MYOG also increased (Figure 1A,C,D). Immunofluorescence staining indicated that MYOC displayed the lowest fluorescence intensity when undifferentiated (0 days). By day 7, the fluorescence intensity of MYOC protein was the highest, and the number and length of myotubes in the field of vision were greatest (Figure 1E). Furthermore, the cells were fixed on day 7 of differentiation, and the laser confocal microscopy results demonstrated that fluorescence due to MYOC protein was strongest at the cell membrane (Figure 1F). The results suggest that the expression of MYOC increased with greater differentiation of C2C12 cells, with MYOC mainly expressed on the cell membrane in highly differentiated C2C12 cells.

### 3.2. Effects of MYOC on the Differentiation of C2C12 Cells

To determine the effect of MYOC on the differentiation of C2C12 cells, pcDNA3.1(+)-MYOC was transfected into C2C12 cells using polyethylenimine (PEI; Sigma, Aldrich, St. Louis, MO, USA). The C2C12 cells were induced to differentiate for 48 h and 72 h. Western blotting indicated that MYOC expression levels increased significantly (Figure 2A,B). Additionally, higher levels of MYH2 and MYOG were expressed after upregulation of MYOC (Figure 2A,C,D). The rate of C2C12 myotube fusion increased from 33.24% to 58.28% (Figure 2E,F). Moreover, after transfection of T3 into the C2C12 cells and differentiated for 48 h and 72 h, the expression levels of MYH2 and MYOG decreased after MYOC expression was silenced (Figure 2G–J), with myotubule fusion rate decreasing from 36.29% to 21.15% (Figure 2K,L). These results indicate that MYOC influenced the differentiation of C2C12 cells.

### 3.3. Interaction and Co-Localization of MYOC and CAV1

To further explore the mechanisms by which MYOC promoted C2C12 differentiation, immunoprecipitation mass spectrometry sequencing of MYOC was performed to identify the proteins that bound MYOC. The associated mass spectrometry results are displayed in Appendix A, in which the CAV1 protein at its peak value was screened. CAV1 is a broad-spectrum protein and it has been demonstrated in multiple studies to be related to the differentiation of C2C12 cells. Therefore, their relationship was analyzed. The lysates of C2C12 cells on day 7 of differentiation were collected for immune-coprecipitation. The experimental results indicated that when immunoprecipitation was conducted using an anti-MYOC antibody, and the sample was analyzed by Western blotting using an anti-CAV1 antibody, CAV1 protein was identified within the precipitation complex. Similarly, when immunoprecipitated using anti-CAV1 antibody, Western blotting with anti-MYOC antibody also demonstrated MYOC protein within the precipitation complex (Figure 3A). The expression of MYOC and CAV1 proteins in C2C12 cells on day 7 of differentiation was identified by laser confocal microscopy. Fluorescently labeled MYOC and CAV1 were co-located on the cell membrane in a definite pattern of dots (Figure 3B).

### 3.4. Effects of CAV1 on the Differentiation and Localization of C2C12 Cells

To explore the effect of CAV1 on the differentiation of C2C12 cells, we evaluated CAV1 expression levels at different stages of C2C12 cell differentiation. The results demonstrate that the cells gradually expressed higher levels of CAV1 protein as the differentiation of C2C12 cells proceeded. The highest fluorescence intensity of CAV1 protein expression was observed on day 7 of differentiation (Figure 4A,B). Changes in protein expression levels of the differentiation markers MYH2 and MYOG were also consistent with the expression levels of CAV1 protein (Figure 4A,C,D). Additionally, the highest fluorescence intensity of CAV1 was observed on day 7 of cell differentiation (Figure 4E). It was established that CAV1 changed significantly as the C2C12 cells differentiated. Laser confocal microscopy results demonstrated that CAV1 was distributed on the cell membrane (Figure 4F).

To further explore the influence of CAV1 on the differentiation of C2C12 cells, S1 was transfected into C2C12 cells, after which protein samples were collected. The results indicate that when CAV1 expression was inhibited, protein expression levels of MYH2 and MYOG also decreased (Figure 4G–J), as did the rate of fusion of myotubes, decreasing by 21.48% compared with the control group (Figure 4K,L). As demonstrated in previous reports, we demonstrated that CAV1 promoted the differentiation of C2C12 cells [25]. To confirm the relationship between CAV1 and the TGF-β pathway, the phosphorylation levels of Smad2 and Smad4 were measured after decreased CAV1 expression. The results showed that P-Smad2 and P-smad4 expression decreased, establishing that TGF-β activity was inhibited (Figure 4M–P). These results demonstrate that CAV1 was able to regulate the differentiation of C2C12 cells via the TGF-β pathway.

### 3.5. MYOC Exerts an Effect on the Differentiation of C2C12 Cells by Regulation of CAV1 That Affects the TGF-β Pathway

To investigate the relationship between MYOC and the TGF-β pathway, the phosphorylation of Smad2 and Smad4 was measured after MYOC was inhibited. The results indicate that phosphorylated protein expression of Smad2 and Smad4 was inhibited as MYOC was inhibited (Figure 5A–D), demonstrating that MYOC regulated the activity of the TGF-β pathway during the differentiation of C2C12 cells. The results above establish that MYOC binds CAV1 and is important in the differentiation of C2C12 cells, and that both MYOC and CAV1 regulate the differentiation of C2C12 cells via the TGF-β pathway.

To explore the functional relationship between MYOC, CAV1, and the TGF-β pathway, co-transfection methodology was utilized. The results showed when MYOC was activated and CAV1 was inhibited, it was found that the activation effect of MYOC on other molecules was inhibited by CAV1. By detecting the myotube fusion rate and signaling pathway active form of the protein P-smad4, we showed that MYOC activation did not activate the TGF-β pathway when CAV1 was inhibited, and cell differentiation was not altered by MYOC activation. The effect of Desmin immunofluorescence also validated this conclusion. CAV1 was inhibited when MYOC was overexpressed, the ability of MYOC to activate was reduced, and its promotion of the differentiation of C2C12 cells and the activity of the TGF-β pathway was blocked by CAV1. Therefore, we were able to demonstrate that MYOC activated the TGF-β pathway through CAV1 and affects SMAD4 activity, resulting in activation of Smd4 activity in the nucleus, thereby activating transcription factors that regulate differentiation followed by cell differentiation.

## 4. Discussion

MYOC is a 55 to 57 kd secreted glycoprotein belonging to the olfactomedin protein family [26]. A large number of studies have demonstrated that MYOC is expressed at high levels in skeletal muscle cells [27,28,29]. MYOC also plays an important role in skeletal muscle hypertrophy [30,31]. However, the mechanism by which MYOC controls the differentiation of skeletal muscle is poorly reported. Here, we confirm that MYOC expression increased gradually with the fusion of myotubes in myocytes during differentiation, and localized on the cell membrane (Figure 1). C2C12 differentiation was positively regulated when MYOC was overexpressed, whereas it was inhibited when MYOC was inhibited. Therefore, we further confirm that MYOC expression was able to promote C2C12 differentiation (Figure 2).

To explore the mechanism by which MYOC promoted the differentiation of C2C12 cells, the manner in which signals were transmitted from the extracellular to the intracellular environment was explored. The literature suggests that MYOC belongs to the olfactomedin protein family and contains a 30 kDa olfactomedin (OLF) domain. The majority of OLF proteins (including MYOC) are secretory glycoproteins [26]. Structurally, MYOC generally forms a homodimer and high molecular weight aggregates via intermolecular disulfide bridges [32]. According to the above analysis, MYOC is a secreted glycoprotein. In the process of protein secretion, it first lies in the endoplasmic reticulum/Golgi apparatus, but the secreted protein crosses the cell membrane when it is finally secreted outside the cell. So we hypothesized that MYOC could bind to proteins on the membrane in two ways. The first way is that MYOC binds to a protein on the membrane in the process of secretion and the second way is that it is secreted outside of the membrane, then it is attached to the membrane, and binds to the protein on the membrane.

CAV1 belongs to the fossa protein family, an integrator membrane protein that constitutes the cell membrane. Therefore, we believe that CAV1 has the capability to bind to MYOC. Moreover, CAV1 can regulate cellular signal transduction both inside and outside the cell membrane [33]. It was reported that CAV1 can induce the osteogenic differentiation of mesenchymal stem cells (MSCs) [34]. In addition, CAV1 induces the differentiation of C2C12 cells into osteoblasts and adipocytes [25]. This strengthens our judgment that it possesses the ability to promote the differentiation of C2C12 cells. This evidence represents the theoretical basis for the selection of CAV1 from the results of mass spectrometry sequencing for subsequent research. We guessed that the Non-cytoplasmic domain of CAV1 combine with the Non-cytoplasmic domain of MYOC, then come into play. Co-IP was used to verify the interaction of MYOC with CAV1. We used laser confocal microscopy to establish that they have a common expression site in differentiated C2C12 cells. Thus, we speculate that MYOC and CAV1 can combine, thereby playing a role in differentiation (Figure 3). We also demonstrated that CAV1 promoted the differentiation of C2C12 cells (Figure 4).

To explore the manner in which signals from the membrane were transmitted into the cytoplasm, we found from the literature that CAV1 plays an important role in the regulation of TGF-β signal transduction [35]. CAV1 can activate the TGF-β pathway in mouse embryonic fibroblasts and rat myoblasts [18,20]. In previous studies, TGF-β was shown to be a potent inhibitor of myogenesis [16,17]. In recent studies in our lab, we demonstrated that activation of the TGF-β pathway can promote the differentiation of C2C12 cells [36]. Therefore, we believe that CAV1 also affects the differentiation of C2C12 cells by regulation of the TGF-β pathway. In C2C12 cells, the TGF-β pathway is activated, which positively regulates C2C12 cell differentiation. Therefore, we hypothesized that MYOC would be capable of inducing the differentiation of C2C12 cells by affecting the TGF-β pathway through CAV1. After sequential inhibition of MYOC and CAV1, the corresponding decrease in active factors in the TGF-β pathway proved that our conjecture was correct (Figure 4 and Figure 5).

To further establish the relationship between MYOC, CAV1, and the TGF-β pathway, we conducted co-transfection experiments. Inhibition of CAV1 blocked the promoting effect of MYOC activation on C2C12 differentiation (Figure 5). Therefore, we demonstrated that MYOC activates the TGF-β pathway by promoting the expression of CAV1 and thus inducing the differentiation of C2C12 cells.

We further considered the mechanism by which MYOC affects the TGF-β pathway. In our study, MYOC bound to CAV1 and then activated the TGF-β pathway. However, the specific sites affecting the TGF-β pathway are unclear. Studies have shown that CAV1 can bind to Activin receptor-like Kinase 1 (ALK1) in rat myoblasts. ALK1 is in the TβR-I family, the activation of which promotes the TGF-β pathway [18,20]. Therefore, we hypothesized that MYOC can activate AKL1 by activating CAV1 in C2C12 cells that positively regulates the TGF-β pathway. Their relationship will be investigated in more depth in the future.

## 5. Conclusions

The present study is the first to demonstrate the mechanism of action of MYOC in C2C12 cells. MYOC can bind to CAV1 on the membrane and affect the phosphorylation of the TGF-β pathway by activation of CAV1 to induce the differentiation of C2C12 cells. In conclusion, our findings provide a novel method of exploring the mechanism of muscle differentiation and represent a potential novel method for the treatment of muscle diseases.

## Figures and Tables

**Figure 1 biology-10-00686-f001:**
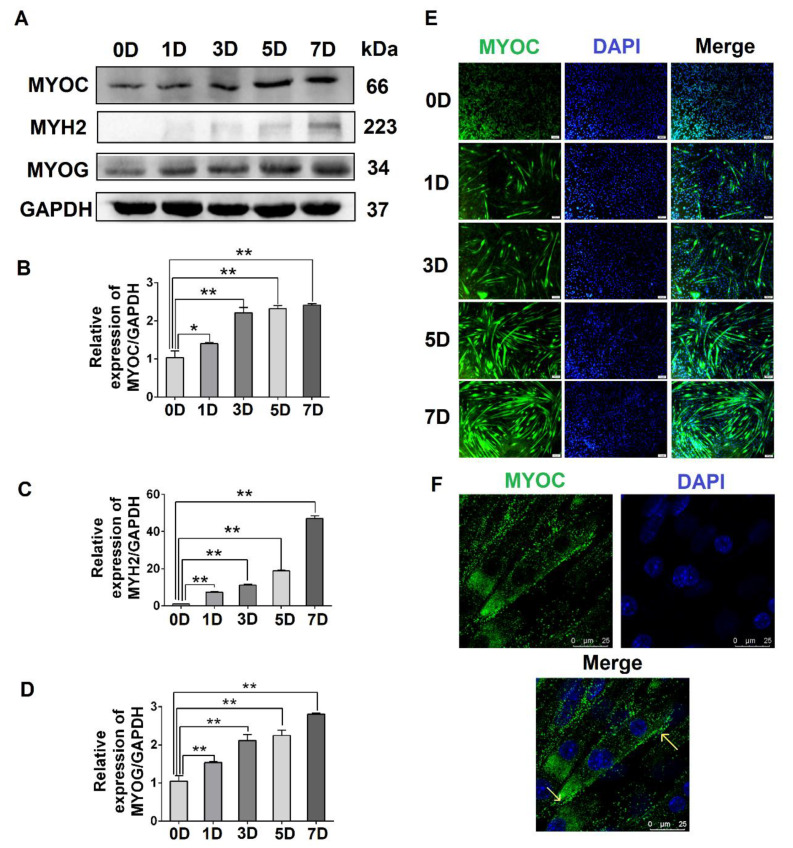
The effect of differentiation on MYOC levels and its localization in C2C12 cells. (**A**). Change in MYOC, MYH2, and MYOG expression levels during C2C12 cell differentiation. (**B**). Gray scanning of MYOC in Figure A. (**C**). Gray-level scanning of MYH2 in Figure A. (**D**). Gray scanning of MYOG in Figure A. (**E**). Immunofluorescence was used to measure changes in MYOC expression and the size and length of myotubes during the differentiation of C2C12 cells. (**F**). MYOC was labeled green with FITC, while the nucleus was stained blue with DAPI. MYOC was principally located on the cell membrane during the differentiation of C2C12 cells, as identified by laser confocal detection. (** *p* values < 0.01, * *p* values < 0.05) (*n* = 3).

**Figure 2 biology-10-00686-f002:**
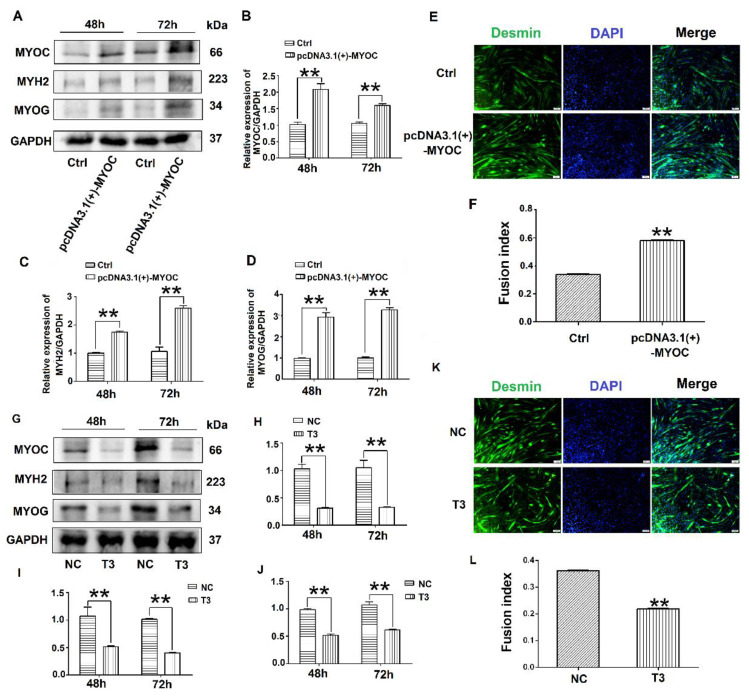
Influence of MYOC overexpression on the inhibition of C2C12 cell differentiation. (**A**). Protein expression of various molecules transfected with pcDNA3.1(+)-MYOC and the empty vector for 48 h and 72 h respectively. (**B**). Grayscale scans of MYOC in Figure A. (**C**). Grayscale scans of MYH2 in Figure A. (**D**), Grayscale scanning of MYOG in Figure A. (**E**). Comparison of fusion rate of myotubes after transfection with empty vector and overexpression vector 5 days later, Desmin stained in green and DAPI stained blue. (**F**). Figure E Statistics of myotubule fusion rate. (**G**). Protein expression of various molecules transfected with NC and T3 fragments for 48 h and 72 h, respectively. (**H**). Grayscale scans of MYOC in Figure G. (**I**). Grayscale scans of MYH2 in Figure G. (**J**). Grayscale scanning of MYOG in Figure G. (**K**). Fusion rate of myotubules 5 days after transfection with NC and T3 sequences. Green represents Desmin staining, blue DAPI staining. (**L**). Statistics of myotubule fusion rate in Figure K. (** *p* values < 0.01) (*n* = 3).

**Figure 3 biology-10-00686-f003:**
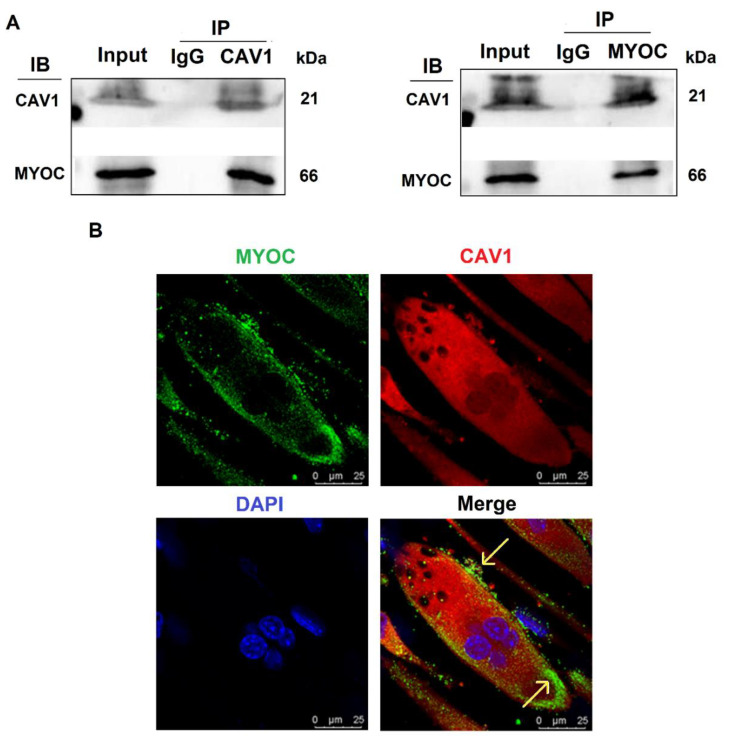
Interaction and localization of MYOC and CAV1. (**A**). MYOC and CAV1 antibodies were used for coprecipitation respectively, each able to be precipitated. (**B**). Location of MYOC and CAV1 in highly differentiated C2C12 cells as observed using laser confocal microscopy. MYOC was labeled as green using FITC-488, CAV1 was labeled red using RBFITC, and the nucleus was stained blue using DAPI.

**Figure 4 biology-10-00686-f004:**
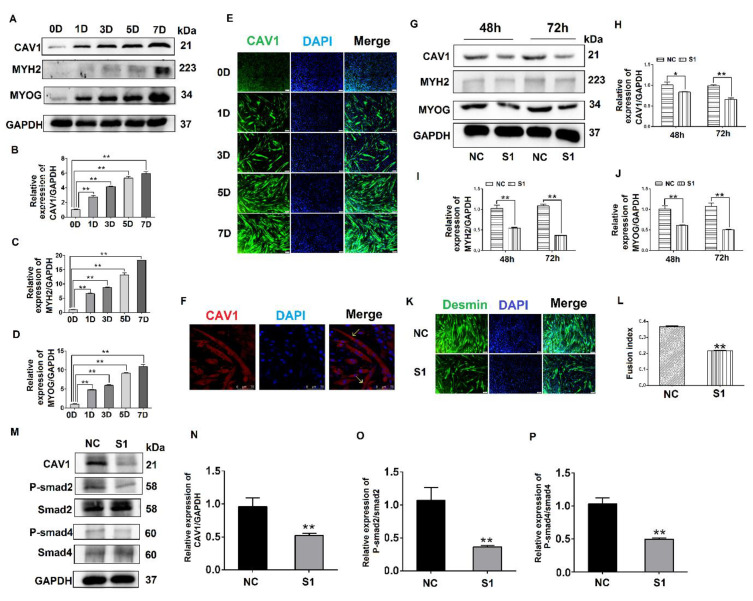
Effects of CAV1 on C2C12 cell differentiation. (**A**). Change in CAV1, MYH2, and MYOG expression levels at different stages of C2C12 cell differentiation. (**B**). Grayscale scan of CAV1 in Figure A. (**C**). Grayscale scan of MYH2 in Figure A. (**D**). Grayscale scan of MYOG in Figure A. (**E**). Immunofluorescence was used to measure the changes in CAV1 expression levels and the morphology of myotubes at various stages of C2C12 differentiation. CAV1 was labeled green using FITC and the nucleus blue using DAPI. (**F**). Laser confocal microscopy demonstrating the location of CAV1 expression, stained red using RBFITC, and nuclei blue using DAPI. (**G**). Changes in CAV1, MYH2, and MYOG expression levels after siRNA transfection for 48 h and 72 h. (**H**). Grayscale scan of CAV1 in Figure G. (**I**). Grayscale scan of MYH2 in Figure G. (**J**). Grayscale scan of MYOG in Figure G. (**K**). Changes in the fusion rate of C2C12 cells in mice after CAV1 inhibition by immunofluorescence detection. (**L**). Myotubule fusion rate in Figure K. (**M**). Change in CAV1, P-Smad2, and P-Smad4 expression after siRNA fragment S1 transfection of CAV1. (**N**). Grayscale scan of CAV1 in Figure M. (**O**). Grayscale scan of P-Smad2 in Figure M. (**P**). Grayscale scan of P-Smad4 in Figure M. (** *p* values < 0.01, * *p* values < 0.05) (*n* = 3).

**Figure 5 biology-10-00686-f005:**
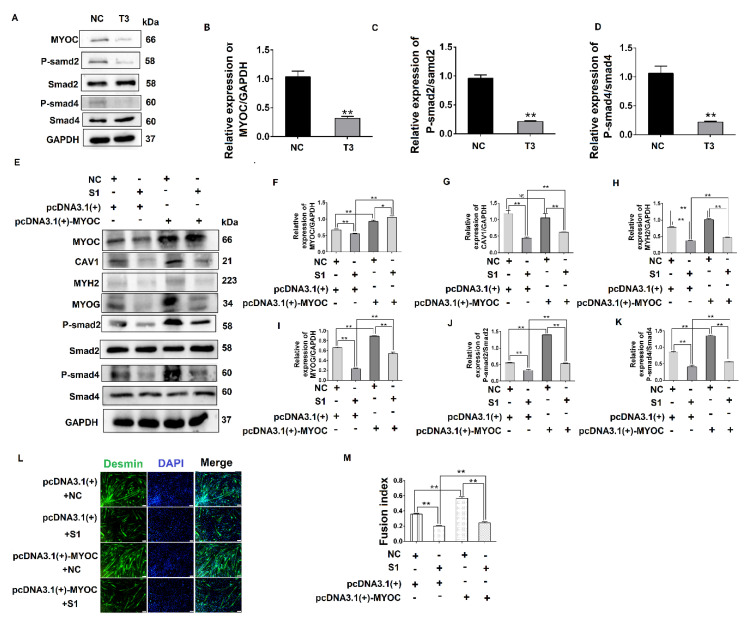
MYOC exerts an effect on the differentiation of C2C12 cells by regulation of CAV1 that affects the TGF-β pathway. (**A**). Change in MYOC, P-Smad2, and P-SMad4 expression after transfection of siRNA-sequence T3 for MYOC silencing. (**B**). Grayscale scan of MYOC in Figure A. (**C**). Grayscale scan of P-Smad2 in Figure A. (**D**). Grayscale scan of P-Smad4 in Figure A. (**E**). Western blotting of all indices in the co-transfection experiment control group and 3 experimental groups. (**F**). Grayscale scan of MYOC in Figure E. (**G**). Grayscale scan of CAV1 in Figure E. (**H**). Grayscale scan of MYH2 in Figure E. (**I**). Grayscale scan of MYOG in Figure E. (**J**). Grayscale scan of P-Smad2 in Figure E. (**K**). Grayscale scanning of P-Smad4 in Figure E. (**L**). Change in rate of fusion of myotubes after co-transfection, as measured by laser confocal microscopy. (**M**). Myotubule fusion rate in Figure L. (** *p* values < 0.01, * *p* values < 0.05) (*n* = 3).

## Data Availability

Not applicable.

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
