# Peer review of "MYOC Promotes the Differentiation of C2C12 Cells by Regulation of the TGF-β Signaling Pathways via CAV1"

_biology, 2021, doi:10.3390/biology10070686_

Round 1

Reviewer 1 Report

Dear authors,

Well done with addressing most of my major and minor suggestions and points.

The last thing, it seems that references 8 and 12 are not well-formatted.

I don't have more comments about the manuscript.

Kind regards,

Author Response

Dear Reviewer

  Thank you indeed for your timely suggestion. I’m really sorry for my carelessness. I have revised the formats of the references and used “Track Changes” function while revising the manuscript so that all the changes can be seen clearly.

Thank you again for your helpful suggestion.

Yours Sincerely,

Yunqin Yan

Reviewer 2 Report

Thank you for the revised version

Author Response

 Dear Reviewer

  I really appreciate the suggestion you have offered us. It’s of great help to us. Thank you again for that.  

Yours Sincerely,

Yunqin Yan

This manuscript is a resubmission of an earlier submission. The following is a list of the peer review reports and author responses from that submission.

Round 1

Reviewer 1 Report

The authors did not substantively change their manuscript in response to numerous critiques including most importantly the request for further exploration of membrane localization, figuring out what part of myocilin is binding to what (membrane, CAV1), and improving the discussion to better place the science in context. Finally, though the request for uncropped was listed as "in supplemental materials", this reviewer could not find this document. Since CAV1 is expressed throughout the cell, the relevance of the finding that it binds to myocilin, and the inferences about cell differentiation, requires additional experiments. 

Reviewer 2 Report

Dear authors,

Thanks for addressing most of my minor and major comments and edits on the manuscript.

In my opinion, the authors must clarify the technical setup during confocal microscopy. This aims to reproduce their results from different groups. One of their major conclusion (the subcellular colocalization of Cav1 with MYOC) is in fact based on confocal microscopy.

What are the settings used in confocal microscopy? z-Stack images whether used or volume of the ROI scanned should also be clarified as well as the objectives (NA, Oil or water, among other parameters) and software used.

Best regards,

Reviewer 3 Report

It is curious that the authors seem to overall agree with the reviewer comments, but are only willing to do text modifications to improve the quality of the paper, while avoiding additional experiments that can give a more clear idea of the situation in a more physiological (in vivo) context.